# Characterization of the COPD Salivary Fingerprint through Surface Enhanced Raman Spectroscopy: A Pilot Study

**DOI:** 10.3390/diagnostics11030508

**Published:** 2021-03-12

**Authors:** Cristiano Carlomagno, Alice Gualerzi, Silvia Picciolini, Francesca Rodà, Paolo Innocente Banfi, Agata Lax, Marzia Bedoni

**Affiliations:** IRCCS Fondazione Don Carlo Gnocchi ONLUS, Via Capecelatro 66, 20148 Milan, Italy; ccarlomagno@dongnocchi.it (C.C.); agualerzi@dongnocchi.it (A.G.); spicciolini@dongnocchi.it (S.P.); froda@dongnocchi.it (F.R.); pabanfi@dongnocchi.it (P.I.B.); alax@dongnocchi.it (A.L.)

**Keywords:** SERS, COPD, multivariate analysis, saliva

## Abstract

Chronic Obstructive Pulmonary Disease (COPD) is a debilitating pathology characterized by reduced lung function, breathlessness and rapid and unrelenting decrease in quality of life. The severity rate and the therapy selection are strictly dependent on various parameters verifiable after years of clinical observations, missing a direct biomarker associated with COPD. In this work, we report the methodological application of Surface Enhanced Raman Spectroscopy combined with Multivariate statistics for the analysis of saliva samples collected from 15 patients affected by COPD and 15 related healthy subjects in a pilot study. The comparative Raman analysis allowed to determine a specific signature of the pathological saliva, highlighting differences in determined biological species, already studied and characterized in COPD onset, compared to the Raman signature of healthy samples. The unsupervised principal component analysis and hierarchical clustering revealed a sharp data dispersion between the two experimental groups. Using the linear discriminant analysis, we created a classification model able to discriminate the collected signals with accuracies, specificities, and sensitivities of more than 98%. The results of this preliminary study are promising for further applications of Raman spectroscopy in the COPD clinical field.

## 1. Introduction

Chronic Obstructive Pulmonary Disease (COPD) is a chronic and unrelenting lung syndrome that causes limitations in physiological air flows, leading to airway remodeling, pulmonary emphysema, and to death in the 20% of the cases, with an incidence between 4% and 10% and smoking habits recognized as one of the principal risk factors [1]. Despite the fast diagnostic procedure that involves the Forced Expiratory Volume in 1 s/Forced Vital Capacity (FEV1/FEV), there are still substantial issues regarding the management of COPD. Some examples are provided by the definition of the COPD phenotypes, which nowadays is performed following a combination of parameters with clinical significance including symptoms, exacerbations, responses to rehabilitation treatments, progression rates or death, in a time-consuming procedure [2]. The exacerbation and hospitalization risks associated with a single patient or to a specific COPD phenotype have not yet been assessed, whereas the therapy effectiveness also relies mainly on the continuous monitoring of patients clinical symptoms during the hospitalizations [1,3]. A correlated critical issue regarding the management of COPD includes the therapy adherence and the personalized respiratory rehabilitation; in fact, treatments effectiveness relies mainly on the characterization of the COPD phenotype and on the adherence to the prescribed therapies [4]. The non-adherence of patients to the continuous therapy regards not only the disadvantages for the affected subjects, but also the important costs for the national health system [5]. Trying to overcome all the listed problems related to COPD, researchers are focused on the identification of a new, fast, and highly informative approach able to identify a biomarker that could allow to evaluate and monitor the therapy and respiratory rehabilitation effectiveness or adherence, and to fully characterize the patients’ biochemical equilibrium in the physiological and pathological state. In recent years, the vibrational Raman Spectroscopy (RS) has been gradually adapted to the characterization of proteins, lipids, nucleic acids, metabolites, and hormones inside specific biofluids with the aim to individuate a specific fingerprint for a pathological onset [6]. RS represents in this frame an ideal methodology due to the rapidity of the analysis, the elevated sensitivity, and the minimal or no sample preparation required. The output signal represents a complex combination of all the concentrations, interactions, modifications, presences, and environments of physiological or pathological biomolecules present in the sample of interest, thus giving a biochemical profile of the sample [7]. The deep analysis of the Raman spectra can not only provide information about the vibrational modes of the molecules, but also about the macro-organization and assembly of the most represented biological species investigated. In many cases, the application of metal nanostructures for the Raman analysis al-lowed to obtain more biochemical information thanks to the enhancement of the signals intensity, due to the Surface Enhanced Raman Scattering (SERS) effect. The SERS effect can be induced by metallic nanostructures such as nanoparticles or by metallic nanostructured materials. Besides gold and silver nanoparticles, aluminum foils demonstrated their potential application in this field due to the detailed SERS signal provided from biological fluids and due to the cheapness of the material [8]. Consecutively, the comparative Raman and SERS investigation is able to characterize the different families (or in specific cases also the single molecules) responsible for the main differences between biofluids collected from healthy or pathological subjects [8]. Due to the signal complexity, analytical procedures have been associated to RS for the reduction of the collected data amount and for the decryption of common trends and differential spectral regions [9]. Multivariate Analysis (MVA), in particular Principal Component Analysis (PCA) and Linear Discriminant Analysis (LDA), allows to reduce the Raman data dimensionality extrapolating informative variables uniquely associated to the relative spectrum, that can be used to evaluate the differences in the signals and to create a classification model [10]. Both Raman and SERS regimen combined with MVA have been already proposed and used for the identification of significant differences in Raman spectra of various pathologies including neurodegenerative diseases, cancers, viral, and bacterial infections analyzing various biofluids among which are blood, serum, plasma, extracellular vesicles, cerebrospinal fluid, saliva, and urine [8,11,12,13,14,15,16,17]. One of the most promising results has been reached using saliva as ideal biofluid, due to the variety of significant molecules contained and to the easiness and repeatability of the collection procedure. Saliva is a complex biofluid with a large pattern of biological molecules shared with the blood stream, some of which have been identified as potential COPD biomarkers, such as C-reactive protein, neutrophil elastases, molecules related to the Radical Oxygen Species (ROS) stress, and procalcitonin [18,19,20,21,22]. In this work, RS in SERS regimen has been used for the analysis of saliva collected from 15 patients affected by COPD and from 15 age- and sex- matched healthy subjects (CTRL). The analytical procedure was optimized taking into consideration two Raman substrates and evaluating the effects of saliva deposition and acquisition procedure. Once optimized, the protocol was adopted for the creation of a Raman database used for the MVA. The final classification model was able to determine the single spectrum membership with accuracy, sensitivity, specificity, and precision of more than 98%. The obtained results represent a proof-of-concept for the potential application of Raman analysis used as diagnostic and monitoring tool in the clinical field.

## 2. Materials and Methods

### 2.1. Materials

All the materials were purchased from Merck KGaA (Merck KGaA, Darmstad, Germany) and used as received, if not specified. Raman-grade calcium fluoride disks (CaF2) were purchased from Crystran LTD (Crystran LTD, Poole, United Kingdom) and used without further purification steps. For the collection of saliva, Salivette^®^ swabs were purchased from Sarstedt (Sarstedt AG & CO, Numbrecht, Germany). Aluminum foil was purchased from Merck KGaA (Merck KGaA, Darmstad, Germany) and used as received to cover the glass substrate for the SERS analysis. All the materials were used following the manufacturer’s instructions.

### 2.2. Patients Selection and Saliva Collection

All the participants provided written informed consent after the approval of the study from the institutional review board at IRCCS Fondazione Don Carlo Gnocchi ONLUS on 11th December 2019. For the proposed pilot study, a total number of 15 COPD patients (n = 15) and 15 CTRL (n = 15) were recruited for this study at IRCCS Fondazione Don Carlo Gnocchi ONLUS, Milan (Italy). COPD patients were recruited with a postbronchodilator ratio of FEV1/FEV < 0.7. The severity of airflow limitation and phenotypes were defined as described by the GOLD grading system [5], including Grade 2, 3, or 4 and subgrades. Exclusion criteria were the combination with obstructive sleep apnea, cancer, minimum state examination < 24, at least 4 weeks from the last acute exacerbation, gingivitis, periodontal diseases, general bleeding of the gum, oral bacterial or/and fungal infections, recent dental operations, and other important comorbidities including cardiovascular, neurologic, and kidney diseases, age < 18. Frequent exacerbators were defined by at least two treated exacerbations per year. Smoking habits were collected and defined as actual smoker or ex-smoker (at least one year without smoking). Sex and age-matched CTRL were recruited. A detailed description of inclusion and exclusion criteria, as well as of the analytical procedure, is reported on ClinicalTrials.gov (ClinicalTrials.gov Identifier: NCT04628962; Title: Raman Analysis of Saliva as Biomarker of COPD). The saliva collection procedure was performed following the instructions provided by Sarstedt. Briefly, a swab was placed in the mouth and chewed for one minute, in order to stimulate salivation. To maintain data comparability, avoiding fluctuations of specific molecules during the day (e.g., cortisol), the collection procedure was performed at fixed time point at least two hours after the last meal and teeth brushing. The collection time was fixed between 3 and 5 pm on Tuesday during the clinical controls at Santa Maria Nascente Hospital (IRCCS Fondazione Don Carlo Gnocchi). Storage time and temperature, participants smoking and particular dietary habits, and time between the collection and the Raman analysis were recorded. Collected samples were stored at −20 °C before the Raman analysis.

### 2.3. Raman Analysis

Before the analysis, saliva samples were towed and centrifuged at 1000× *g* for 5 min in order to recover the biofluid. A drop of saliva (3 µL) was deposited on a CaF2 disk or on a commercially available aluminum foil to provide the Raman signal enhancement (SERS) thanks to its metallic nature and to the micro- nano-structure, slightly modifying our previous protocol [8]. In this work, the filtration step using filters with different cut-offs was avoided, in order to keep the physiological and pathological amount of biological molecules inside saliva. After the deposition step, saliva was dried at room temperature. The Raman analysis was performed using a Raman microscope Aramis (Horiba Jobin-Yvon, France), equipped with a 785 nm laser source emitting at 512 mW. The silicon reference band at 520.7 cm^−1^ was used as reference for the calibration procedure. For all the analysis, a 50× objective (Olympus, Japan) was used, with acquisition time of 30 s, grating at 600 and hole at 400. The spectral range was set between 400 and 1600 cm^−1^ while the range was restricted to 400–1500 cm^−1^ for the analysis on calcium fluoride. The final spectral resolution was 0.8 cm^−1^/step. For each subject involved in the study, the Raman acquisition was performed following a square-map with at least 25 points (60 µm × 80 µm) or, in the CaF2 cases, focalizing all the points on the interest area. All the methods and procedures described in this work were performed in accordance with the relevant guidelines and regulations.

### 2.4. Data Processing and Statistical Analysis

Before the MVA process, all the spectra were pre-processed in order to uniform and homogenize the Raman dataset. In particular, the raw acquired data were fitted with a fifth-degree polynomial baseline, using 79 points to interpolate the baseline and 21 points to determine the noise. All the spectra were resized on the reference band at 1001.5 cm^−1^ and smoothed using a second-degree Savitzky–Golay method in order to remove non-informative spikes. Data were extracted with a final resolution of 0.98 cm^−1^/step, acquiring 985 points for single spectrum. The contribution of the aluminum substrate was removed from all the spectra acquired on the same substrate. Artifact spectra due to fluorescence or Raman *z*-axis de-focus, were removed resulting in a 468 spectral set. Once pre-processed, the spectra collected from the two experimental groups were used as means to show the average signal obtained from COPD and CTRL. LDA was performed in order to assess a preliminary difference between the groups, reducing data dimensionality extracting and characterizing the Principal Components (PCs) with the highest loading. The first 15 PCs (cumulative loading 86.28%) were used to create the LDA-based classification model avoiding the classification overfitting and extracting the informative Canonical Variables (CVs) used for the Leave-One Out Cross-Validation (LOOCV), hierarchical clustering, and related confusion matrix. LOOCV was applied on the single spectrum collected, not considering the spectral pattern associated to the subject, in order to verify the effective differences between the Raman datasets. The Receiver Operating Characteristic (ROC) curve was calculated using the sensitivities and specificities associated to the confusion matrix, as described in literature [23], reporting the obtained Area Under the Curve (AUC). Matthews Correlation Coefficient (MCC) was calculated in order to assess the quality of the binary classification model. ANOVA test was used to assess the statistical relevant differences between the analyzed groups. All the statistical tests were performed using OriginPro 2018 (OriginLab, Northampton, version 2021, USA) and MedCalc (MedCalc Software, Ostend, version14.8.1., Belgium).

## 3. Results

### 3.1. SERS Methodology

The Raman analysis of saliva was performed optimizing the analytical protocol developed by our group, reporting the effects of the different acquisition parameters [8]. The previous selective filtering procedure (cut-off 3kDa) was avoided in order to (i) decrease the time for sample preparation, (ii) make the analysis cheaper, and, more importantly, (iii) to preserve the original biochemical pattern of molecules inside the biofluid. The first analysis was performed on a drop of saliva (3 µL), dried at room temperature, and deposited on a CaF2 disk. CaF2 is a Raman standard substrate, chosen for its negligible Raman signal. After the drying procedure, all the biomaterials present in saliva create regular aggregates as a result of the evaporating concentration gradient affected by the presence of organic mucin matrices (Figure 1A) [24,25]. The Raman spectra were collected on the volumetric deposition, obtaining a detailed and intense Raman signal respect to the spot on the drop plane. Figure 1B shows the Raman spectrum collected from the volume deposition in dried saliva samples. The spectrum shows all the characteristic salivary peaks: 441, 517, 599, 715, 750, 866, 920, 978, 1001, 1047, 1157, 1203, 1244, 1268, 1346, and 1444 cm^−1^. The most important signal attribution regards the peak at 750 cm^−1^ related to the O-O stretching vibration in oxygenated proteins. The peaks at 866 and 1157 cm^−1^ can be attributed to the C-N stretching and to the CH3 rocking in protein backbone, respectively, while the peak at 1001 cm^−1^ is related to the ring breathing of aromatic amino acids and the signal at 1444 cm^−1^ can be assigned to the C-H stretching of glycoproteins, mostly obtained from mucines [24].

Despite the intense and detailed Raman signal, analysis on CaF2 presents some limitations including the cost of the material and the formation of the double structure after the drying procedure (spot on the drop plane and volume deposition). This last phenomenon highly influences the collection of a repeatable spectrum, generating high variability (Figure 1B, standard deviation), favoring the presence of artefact spectra and providing a low-intensity signal. In order to overcome these limitations, commercially available aluminum was used as Raman substrate for the saliva analysis, modifying the procedure reported by Muro et al. [26]. After the drying step on aluminum, the saliva drop presents a remarked edge, which separates the substrate from the sample. The saliva drop observed with the optical microscope presents a homogeneous surface without volume deposits and spots on the drop plane (Figure 2A, in box), with the Raman signal of a comparable intensity all over the sample and with a lower number of artefact spectra. The signals collected from each region are highly homogenous, as indicated by the low standard deviation associated to the spectrum (Figure 2A). The identified peaks normally correspond to the signals collected using calcium fluoride as substrates, with the most intense bands at 441, 524, 543, 587, 621, 715, 746, 778, 812, 924, 1001, 1051, 1126, 1161, 1267, 1284, 1301, 1382, 1409, and 1454 cm^−1^ (Figure 2A, black arrows). The slight changes in peaks shift and intensities are due to the SERS induced by the metallic substrate provided by the used SERS inducer, with results comparable with the other present in literature about the SERS enhancing [26]. The comparison between the spectra obtained using the two different substrates is shown in Figure 2B. The attribution of the peaks, based on previous studies on saliva and on biological tissues, is reported in Table 1 [7,24,26].

Most of the identified peaks are related to the protein content of saliva (specific amino acids, protein backbone, and secondary structure), lipids carbon signals, nucleotide modes, and glucose or glycogen (Table 1). The abundance of information is related to the high concentration of biomolecules inside the biofluid, and to the strength of the Raman signal provided. Proteins and lipids possess strong Raman effects, deeply described in literature, allowing the precise attribution of the peaks. Examples are provided by the Amide III band that represents the amount and percentage of C-N stretch and N-H bend at 1267 cm^−1^, and by the strong signal of aromatic amino acids such as tryptophan and phenylalanine (Table 1). Depending on the position of the Amide III identified band, the Raman spectra can provide information regarding the secondary structure of the most abundant protein. In this case, the position at 1267 cm^−1^ indicates a most prominent α-helix conformation [27]. Combined with the information collected about the aromatic amino acids, the partial identification of the most abundant group of proteins can be done on the base of the secondary structure and the percentage of specific peptides. Similarly, peaks at specific positions attributed to different molecules can provide information regarding the species involved (Table 1). Several biological molecules have been already attributed to specific Raman signals, which can be altered in specific physiological or pathological conditions, contributing to the identification of alterations in metabolic pathways, damaged products, and metabolic pathways. Phosphatidylserine and phosphatidylinositol are membrane lipids involved in several communication and metabolic pathways, usually interconnected in the protein transportation. The concentration and modifications of these molecules have been connected to the onset of different pathologies including diabetes, cancer, cognitive impairment, and lung disorders such as COPD [28,29,30,31]. Other important information can be deduced from the position of specific peaks attributed to lipids and nucleic acids damaged of differentially expressed because of the damages produced by ROS in stress conditions, specifically [24,32,33]. The Raman signals provided by saccharides, mostly by glucose and glycogen, are indicators of the different accumulation and metabolism inside the body. These markers have been associated to different pathologies related to the altered glucose metabolism, release, and accumulation. Besides diabetes, in COPD onset, the levels and fluctuations of glucose and glycogen have also been associated to different pathological events [34,35]. Considering the increased homogeneity and quality of the Raman spectra of the saliva samples analyzed on the aluminum compared to the difficulty on calcium fluoride, aluminum was selected as substrate for further analysis of clinical samples.

### 3.2. Clinical Analysis

Subjects included in the study were 15 COPD patients (n = 15) and 15 CTRL (n = 15) with the clinical and demographic characteristics reported in Table 2. Number and demographic data of the two experimental groups were comparable, with a good distribution of the GOLD classification among the COPD patients. The individuated COPD phenotypes were classified as COPD with emphysema (n = 7) and COPD with bronchitis (n = 8), without any patient affected by overlapped COPD/Asthma and avoiding the potential signal contamination by other comorbidities.

Taking advantage of the fast analytical Raman procedure and of the large amount of information provided by the salivary spectral analysis, the optimized protocol was adopted for the analysis of saliva collected from 15 patients affected by COPD and 15 CTRL. The aim of the procedure was to ascertain and determine the main differences be-tween the two experimental groups, laying the foundations for the identification of a specific COPD salivary fingerprint. In Figure 3, the average Raman spectra collected from the two groups are presented. The COPD average spectrum presents a uniform shape, with the lower values of calculated standard deviations respect to the CTRL (Figure 3). The peaks attribution reflect the values presented in Table 1, with slight shifts always included between ±4 cm^−1^. The tapered curve collected from COPD indicates a homogeneous spectral trend between the analyzed subjects, promoting the conception of a salivary Raman COPD signature, able to identify the pathological onset.

On the other hand, the signals collected from the CTRL group present a more jagged shape indicating, together with the higher standard deviation values, the disparate distribution of the collected spectra. The main differences comparing the two groups are presented in Figure 4. The overlap of the two average spectra revealed regions of particular interest, in which the signal was different in terms of intensities, peak positions, and presence (Figure 4A, grey boxes). Regions between 500 and 600 cm^−1^, normally due to the signals of lipids and carbohydrates, were more intense and structured in CTRL group with respect to the COPD counterpart. Similarly, the region between 1250 and 1350 cm^−1^ was more prominent in CTRL. As a result of the uniform shape in COPD spectra, only regions between 900 and 950 cm^−1^ and 1100 and 1200 cm^−1^ were more prominent in the COPD group with respect to the CTRL. These two regions were attributed principally to proteins, carbohydrates, and nucleotides vibrational modes. A deeper analysis was performed subtracting the two averages spectra and calculating the associated error propagation, identifying in this way the peaks and bands responsible for the differences between the two experimental groups (Figure 4B). The individuated peaks, belonging to the class in which they were more abundant, are presented in Table 3.

After a general overview, signals related to lipids and protein were more abundant in COPD with respect to the CTRL group, where the saccharides part was more concentrated (Figure 4B and Table 3). In particular, the signals collected from phosphatidylserine and phosphatidylinositol were more intense in CTRL respect to the COPD. The general alteration and loss of lipids, in particular phospholipids, has been widely demonstrated after COPD onset [36]. Availability of phospholipids, in particular phosphatidylserine and phosphatidylinositol, strongly correlates with pulmonary function, consecutively inducing a loss of phosphorylated lipids during the pathological onset [37]. Peaks related to saccharides, including glucose and glycogen, are more represented in the COPD average spectrum (Figure 4B, Table 3) with respect to the CTRL group, probably indicating an accumulation of the molecules. The described scenario reflects the metabolic alterations occurring in COPD onset. Acute hyperglycemia is associated with poor outcomes for different acute and chronic diseases including COPD, where progressive insulin-resistance, altered glucose metabolism, and glucose-mediated hormones responses have been deeply characterized [38,39]. Moreover, plasmatic glucose levels have been proposed as potential biomarkers for the definition and prediction of the exacerbation events in COPD [34]. The hallmarks of protein in the subtraction spectra present in COPD reveal a similar situation, in which the signals related to the single aromatic amino acids and to the vibrational mode of the secondary structure are more significant in the pathological state (Figure 4B, Table 3). The chronic inflammatory and oxidative stressful state induced by COPD generates an overexpression and consecutively a higher presence of specific molecules circulating in different biofluids (e.g., blood, saliva). These molecules are related to the inflammatory system and to the product of the inflammatory response. Different proteins have been characterized in saliva, allowing to discriminate the saliva collected from COPD from the one collected from healthy controls with their higher expression patterns. An example is provided by the work of Patel et al. [40] where higher levels of C-reactive protein, procalcitonin, and neutrophil elastase in saliva were able to highly discriminate the biofluid collected from COPD patients respect to the CTRL group. The same scenario has been characterized in our results, with the contributions of a higher methodological sensitivity provided by RS (up to 10–15 M in SERS regimen) and of the concomitant detection of multiple molecules respect to the ELISA method that can determine the differences between the two groups. Several peaks related to the signal provided by lipids and nucleic acids can be attributed to the damages in the biological structures caused by ROS, highlighted in the COPD spectrum respect to the CTRL (Figure 4B, Table 3). Especially during the continuous stress situations, such as chronic pathologies, the damages due to the ROS increase the products related to this pathway [19]. During the COPD onset and progression, the ROS-related molecules, including lipids and nucleic acids, assume a defined expression pattern allowing the identification of the pathological state [41]. The global overview provided by the grouped analysis of saliva could be able to generate specific information about the biological species involved at the onset of the pathology and during the progression and, once completely interpreted, also revealing useful information regarding the pathological mechanisms and the targets of new therapies. These results can be used to identify a Raman salivary fingerprint that could be used to assess the pathological onset, the identification of different phenotypes, the effects of the prescribed therapies, and to monitor the respiratory rehabilitation efficacy.

### 3.3. Multivariate Analysis and Classification Model

The differences and the particular spectral patterns detected by means of the RS in the COPD and CTRL groups were computed using MVA in order to reduce the volume of data and to extract coefficients that are able to maximize the variance between each spectrum. The PCA is an unsupervised data transformation procedure of complex data that reduces the input variables in a set of independent and orthogonal PCs maximizing the variance [9]. The application of the LDA model on the PCA leads to the identification of the discriminant axes that optimally classify the extracted data on the base of their relationships. As result, the CVs are the coefficients that best describe the optimal data dispersion for potential differences. Consecutively, once the effective different dispersions of PCs and CVs was tested, a LOOCV-based classification model was performed assessing the capability of the Raman signature to be exploited as discriminant marker for the COPD.

Figure 5 shows the results obtained from the PCA procedure. The cumulative loading obtained from the first three extracted PCs was 60.3%, identifying a great part of the detected differences in these data. The areas concerning the different PCs interest all the spectra, with particular regions of dispersion, which highlight the relationship between the variables in the areas of the three considered PCs (Figure 5A). The 3-D distribution of PC1, PC2 and PC3 demonstrated two defined spatial groups clearly indicating the difference between the data (Figure 5B). The uniform shape of the average COPD Raman spectra (Figure 3A) is translated in the more centered dispersion of the PCs collected from the COPD group (Figure 3B). The statistical test performed regarding the influence of the smoking habits on the collected data, revealed a statistical difference (Mann–Whitney test, *p* < 0.05, Data not shown) between the subjects without smoking habits and with past smoking habits. Despite the weak evidence individuated, an increase in subjects’ numerosity will probably decrease the difference encountered due also to the missing statistical differences between subjects with smoking habits and without smoking habits. Similarly, considering the Raman data processed through PCA, the analysis based on the patients severity stages (Table 2) was not performed due to the explorative character of the study and due to the almost single presence of patient in each clinical stage. After the PCA, LDA was performed in order to extract the coefficients for the creation of the classification model. Considering that the LDA is a supervised method, CVs were firstly processed through unsupervised hierarchical clustering, assessing the effective grouping of the collected variable without the labelling procedure (Figure 6A). The resulting dendogram shows the automatic aggregation of the CVs into two groups (COPD and CTRL) at a relatively short distance, confirming the effective belonging to two different classes. The statistical analysis performed on the same data revealed a significant difference between the CVs dispersions of COPD and CTRL (Figure 6B, *p* < 0.001). The further analysis of the CVs dispersion revealed no overlapping between the two groups, with no points indicated as outliers (Figure 6A). After the validation of the data used to create the classification model, LOOCV was applied with the purpose to train the algorithm with the detected differences leading to the independent identification and attribution of each single spectrum collected during the Raman analysis. Each single spectrum was used for the LOOCV at the spectral level, in order to verify the existing differences between the collected datasets, avoiding the single patient labelling. The application of the LOOCV model to the spectral dataset is aimed at the identification of repetitive and constant spectral variations for the creation of a single-spectrum classification model. Despite the identified significant differences, a LOOCV model based on the total spectral pattern of each subject involved in this study (patient-level classification model) cannot be adopted due to the experimental groups’ numerosity. Our scope was to set up a proof-of-concept with the proposed preliminary study, investigating if the analyzed spectral differences can be applied for the creation of a classification model, confirming the spectral difference constancy. Further patients’ enrolment are needed for the definitive validation of the technique at the patient level. The performances of the classification model are reported in Table 4.

Accuracy, sensitivity, specificity, and precision were all equal or more than 98% due to the Error-Rate for cross validation of training data (ER) of 0.85% indicating a minimal error in the spectral attribution process (Table 4). MCC is an indicator used to assess the reliability and the quality of a binary classification calculated on the values of true and false positives and negatives. The obtained value of 0.97 confirmed the ability of the system. As further confirmation, we analyzed the ROC curve, estimating the AUC to 0.975 with a confidence interval of the 95% (Table 4). All the data regarding the performances of the classification model, based on the single Raman spectrum, confirm the ability of the RS to accurately discriminate the signal collected from COPD respect to the CTRL. This finding was firstly highlighted by the different shape of the average Raman spectra collected from the experimental group and definitively confirmed by the deeper MVA. Despite different studies reported in literature about the characterization of several potential circulating biomarkers for the COPD, the main limitations regarding these studies concern the individuation of one, or few, molecules each time using single molecule detection techniques such as ELISA, mass spectroscopy, or nucleic acids amplification [18,29,42,43].

The main advantage of RS regards the detection of the entire biomolecular pattern present in a specific biofluid, treating the entire informative spectrum as whole biomarker. In this way, it is possible to collect information regarding the content and quality of circulating proteins, molecules of the inflammatory system, products of the ROS pathways, structural and signaling lipids, carbohydrates, and alterations in the nucleic acids content. The complexity of the obtained signal combined with the high data volume require a computational technique to deeply investigate the extractable information. The application of MVA to the Raman database has been already proposed and used for almost all the techniques involving RS as diagnostic or body fluids investigative tool [9]. Besides the accuracy and sensitivity, RS is taken into high consideration also for the velocity of the test and for the minimal sample preparation required before the analysis. In the proposed study, the entire analytical process, starting from the collection of saliva to the obtaining of the final result after the MVA, lasts approximately 30 min, with different steps that can be easily reduced (e.g., drying procedure, acquisition time, number of acquired spectra for each patient). Despite the limited number of participants involved in the study, the results in terms of Raman profiles and discriminant power of the final classification model were extremely encouraging, confirming the potential of the proposed methodology. A larger cohort of patients will be fundamental for the standardization of the procedure, introducing more experimental groups in order to characterize the slight changes in Raman spectral structures that define, for example, the patients that completely adhere to the therapy, the risk of exacerbation associated to the single patient, or the different COPD phenotypes for a fast phenotypic identification. This last point is a critical issue in COPD management, nowadays being the identification of phenotypes time-consuming, expensive, and not accurate [2]. The identification of the COPD phenotype involves the clinical observation of the patients, reporting the combination of exacerbation events, respiratory afflictions, other related symptoms, responses to the pharmacological and rehabilitative therapies, and to the survival rate of the patient in a time-consuming (2/3 years) process [2]. Moreover, similar to other pathologies, each phenotype responds in a different way to the prescribed pharmacological and rehabilitative therapies, without a quantifiable biomarker that is able to indicate the optimal route. Once well characterized and validated with more patients and more experimental groups treated with different therapies and analyzed in a transversal and longitudinal study, the RS-based approach could provide a fast and sensitive investigative tool for the clinicians, helping the management of a complex and chronic respiratory pathology. The creation of a Leave-One Patient Out Cross-Validation-based classification model will be of crucial importance to assess the feasibility of the diagnostic and monitoring Raman platform.

## 4. Conclusions

In conclusion, in this pilot study, we proposed the characterization of the COPD salivary Raman fingerprint. Saliva was selected as investigative biofluid due to the highly informative biochemical composition and due to the minimally invasive and easy collection procedure. We evaluated different acquisition parameters taking into consideration both the details and information of the final spectrum and the optimization of an easy-to-acquire and not expensive sample preparation. The analysis on the average spectrum led to the potential attribution of the main biochemical species responsible for the differences between the COPD and CTRL groups, identifying specific protein families, lipids, and saccharides in common with previous studies reporting modifications in the biochemical equilibrium after COPD onset. The consecutive MVA allowed the creation of a classification model able to discriminate the single Raman signal collected from the two experimental groups with accuracy, sensitivity, specificity, and precision of more than 98%. Due to the low number of subjects involved in this preliminary and pilot study, more patients, as well as more healthy subjects and the recruitment of a comparable pathological control group, must be included to definitively assess the application of the RS as diagnostic and monitoring tool in COPD at the patient level. Our preliminary results demonstrated the potentiality of a Raman-based approach that could be use in future not only to assess the COPD pathological onset, but also to identify different phenotypes, evaluate the effects of the prescribed therapies, and to monitor the respiratory rehabilitation efficacy starting from an easily collectable biological fluid.

## Figures and Tables

**Figure 1 diagnostics-11-00508-f001:**
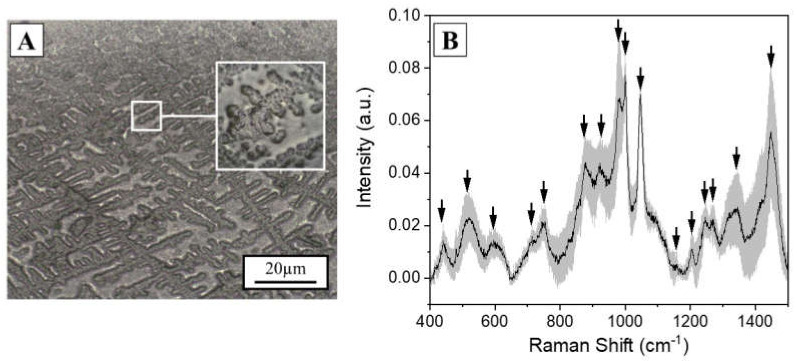
(**A**) Optical microscopy image (10×) of a saliva drop (3µL) dried at room temperature on a Calcium Fluoride disk. Scale bar 20 µm. In box, magnification (50×) of the volumetric mass after the drying procedure. (**B**) Average Raman spectrum of the saliva sample collected on the volumetric mass. The grey band represents the standard deviation. The black arrows indicate the most prominent peaks of interest.

**Figure 2 diagnostics-11-00508-f002:**
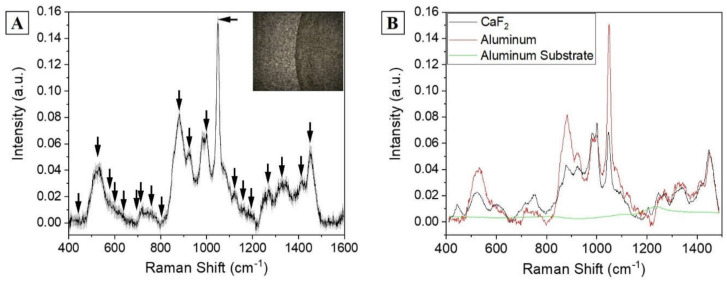
(**A**) Raman average spectra of a drop of saliva dried on aluminum substrate. The grey band represents the standard deviation. The box shows the optical microscopic image of the sample (objective 50×). The black arrows indicate the attributed peaks. (**B**) Comparison between the salivary spectra obtained using CaF2 and aluminum as Raman substrates and the signal of the aluminum substrate without saliva (green line).

**Figure 3 diagnostics-11-00508-f003:**
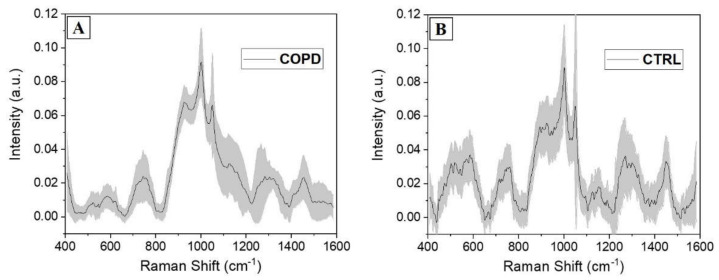
Average salivary Raman spectra collected from the experimental (**A**) COPD and (**B**) CTRL groups. The grey bands represent the standard deviations.

**Figure 4 diagnostics-11-00508-f004:**
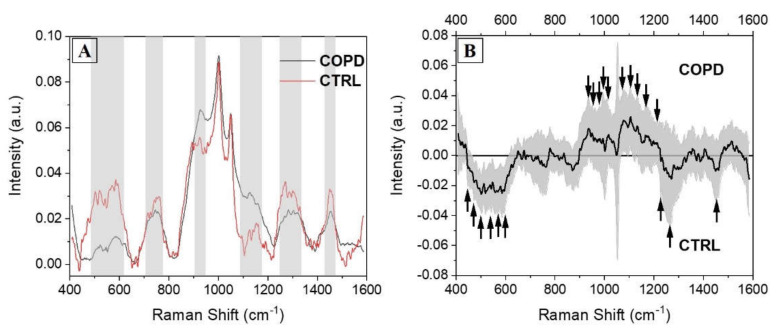
(**A**) Overlapped average Raman spectra collected from COPD (black) and CTRL (red) subjects, with the indications of the regions with the main differences (grey boxes). (**B**) Subtraction spectrum between COPD and CTRL groups. The grey band represents the error propagation calculated from the spectral standard deviations. The black arrows indicate the most important peaks.

**Figure 5 diagnostics-11-00508-f005:**
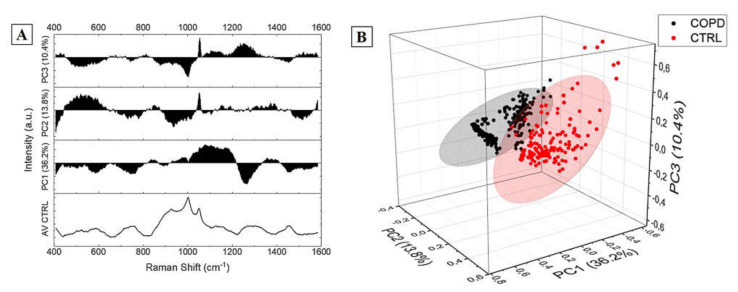
Results of the Principal Component Analysis showing (**A**) the loadings and (**B**) the 3-D distribution with the 95% confidence ellipse of the Principal Component 1 (36.2%), Principal Component 2 (13.8%), and Principal Component 3 (10.4%) cumulatively representing the 60.3% of the components.

**Figure 6 diagnostics-11-00508-f006:**
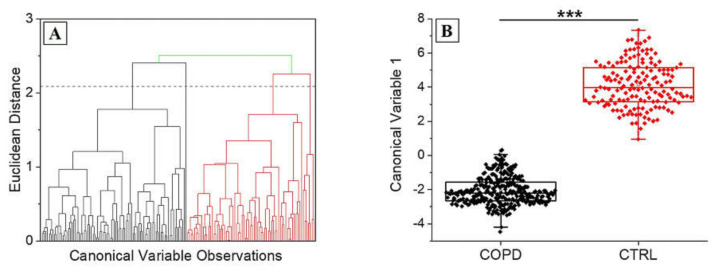
(**A**) Hierarchical clustering dendogram representing the unsupervised grouping of the Canonical Variables on the base of the Euclidean distance. The dashed line represents the convergence distance. (**B**) Dispersion of the Canonical Variable 1 for COPD and CTRL. *** *p* < 0.001 One-Way ANOVA test.

**Table 1 diagnostics-11-00508-t001:** Attribution of the most prominent peaks obtained from Raman salivary analysis (± 4 cm-1), based on reported literature [7,24,26].

Raman Shift	Attribution			
	Protein	Lipids	Nucleotides	Carbohydrates
441 cm^−1^		Sterols stretching		
524 cm^−1^		Phosphatidylserine		
543 cm^−1^				Glucose/Saccharides
587 cm^−1^		Phosphatidylinositol		
621 cm^−1^	Phenylalanine			
715 cm^−1^		C-N phospholipids		
746 cm^−1^			Ring breathing DNA/RNA	
778 cm^−1^			Ring breathing C, U, T	
812 cm^−1^			Phosphodiester bonds	
924 cm^−1^				Glucose/Glycogen
1001 cm^−1^	Phenylalanine, Tryptophan			
1051 cm^−1^				Glycogen
1126 cm^−1^			Stretching of acyl backbone	
1161 cm^−1^	Tyrosine			
1267 cm^−1^	Amide III			
1284 cm^−1^	C-H bending			
1301 cm^−1^			C-H vibration	
1382 cm^−1^	C-H rocking			
1409 cm^−1^	Bending of methyl groups			
1454 cm^−1^		Phospholipids		

**Table 2 diagnostics-11-00508-t002:** Demographic characteristics of the subjects involved in the study. Data are presented as average with the standard deviation (± SD) or percentages (n %) and with the two-sided *p*—Values (p). Smoking habits are defined as actual smokers (Yes), never smoked (No) or previous smoking habits before the last year (Ex). COPD phenotypes and GOLD classification were attributed following [5]. Frequent exacerbators presented at least two treated exacerbations per year. Differences between the groups were analyzed using Chi-square test and Fisher Exact test.

	COPD	CTRL	*p*-Value
**Number**	15	15	−
**Sex (male)**	53.3% (8)	53.3% (8)	1.23
**Age (years)**	66 ± 10	60 ± 6	0.06
**Smoker**	Yes = 46.6% (7) Ex = 54.4% (8)	Yes = 33.3% (5) No = 66.7% (10)	0.27
**COPD Phenotype**	With Emphysema = 46.6% (7) With Bronchitis = 54.4% (8)	−	0.59
**Frequent exacerbator**	Yes = 60% (9) No = 40% (6)	−	0.12
**GOLD Classification**	2 A = 6.6% (1) 2 B = 20% (3) 2 C = 6.6% (1) 2 D = 6.6% (1) 3 C = 6.6% (1) 3 D = 26.6% (4) 4 B = 13.3% (2) 4 D = 13.3% (2)	−	−

**Table 3 diagnostics-11-00508-t003:** Attribution of the principal Raman peaks due to the differences in the subtraction spectrum, following the attribution of [7,24,26]. The Raman shifts are presented under CTRL or COPD depending on the abundance (±0.005 ∆I) in the considered group.

Raman Shift CTRL	Attribution	Raman Shift COPD
441 cm^−1^	Sterols stretching	−
471 cm^−1^	Polysaccharides	−
499 cm^−1^	Glycogen	−
524 cm^−1^	Phosphatidylserine	−
543 cm^−1^	Glucose/Saccharides	−
560 cm^−1^	Tryptophan	−
587 cm^−1^	Phosphatidylinositol	−
−	Glucose/Glycogen	924 cm^−1^
−	Proline	937 cm^−1^
−	Proline and Valine	948 cm^−1^
−	Phosphate monoester groups	962 cm^−1^
−	C-H bending in lipids	979 cm^−1^
−	Phenylalanine, Tryptophan	1001 cm^−1^
−	Glycogen	1051 cm^−1^
−	C-C of lipids	1077 cm^−1^
−	Phenylalanine	1104 cm^−1^
−	Stretching of acyl backbone	1126 cm^−1^
−	Tyrosine	1161 cm^−1^
−	Nucleotides breathing	1195 cm^−1^
1242 cm^−1^	Amide III	−
1267 cm^−1^	Amide III/Lipids	−
1450 cm^−1^	C-H deformations in lipids	−
−	Cytosine	1515 cm^−1^

**Table 4 diagnostics-11-00508-t004:** Results of the model based on Leave-One out Cross-Validation for the assignment of the single spectra to the experimental group. Error-Rate for cross-validation of training data (ER); Matthews Correlation Coefficient (MCC); Receiver Operating Characteristic Area Under the Curve (ROC-AUC).

	Accuracy	Sensitivity	Specificity	Precision	ER	MCC	ROC-AUC
**LOOCV Model**	98%	98%	99%	98%	0.85%	0.97	0.975

## Data Availability

All the data can be obtain after reasonable requests, contacting C.C. or M.B. Information regarding the study can be also found at ClinicalTrials.gov; Identifier: NCT04628962; Title: Raman Analysis of Saliva as Biomarker of COPD.

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
