# Peer review of "Characterization of the COPD Salivary Fingerprint through Surface Enhanced Raman Spectroscopy: A Pilot Study"

_diagnostics, 2021, doi:10.3390/diagnostics11030508_

Round 1

Reviewer 1 Report

The paper "Characterization of the COPd..." by Carlomagno et al is very interesting, original, well written and well organized. 

The only observation that I point out to the authors is that since the saliva samples were collected from a limited number of patients (15 ctrl and 15 copd), the investigation has a pilot study level that requires a greater extension to include a much larger patient population. The SERS technique can be easily scalable to more numerous samples. Therefore, I ask that the pilot study character of the investigation presented be remarked in the abstract, in the conclusions, and possibly also in the title of the article. 

Author Response

We thank the reviewer for the suggestions,

you will find the detailed answer to your comment in the attached file. We changed the manuscript parts indicated by the reviewer, highlighting in red the new version in the revised manuscript (including the new proposed title).

Best regards 

Reviewer 2 Report

Recommend: Minor revision

The author reported the method of SERS to analyze the characteristics of Raman fingerprint in the saliva of COPD patients, and found out the potential attribution of the main biological species that caused the difference between COPD group and CTRL group. The accuracy, specificity and sensitivity of the signal were analyzed by linear discriminant analysis. There are some issues should be clearly solved as follows before its possible acceptance in the Diagnostics.

  1. All the Figures need to be improved. The fonts in the Figures are too small to read.
  2. Some important peaks in Raman spectroscopy should be labeled.

Author Response

We thank the reviewer for the suggestions and corrections aimed to clarify our work.

Attached you will find the detailed answers to your questions, with the list of modifications applied to the new images. In the attached revised manuscript you will find the new images.

Best Regards

Reviewer 3 Report

The article is an original study that undoubtedly contributes to the field of application of spectroscopic methods for the study of saliva. however, from a methodological point of view, I have a number of questions. 1. The sample size is insufficient 15/15 given that the COPD group is very heterogeneous. Patients with varying degrees of severity of COPD are present in almost a single instance. At the same time, given such significant differences between patients with COPD from healthy ones, it is not at all discussed whether patients with moderate and severe COPD differ. Groups should be expanded to at least 5 people of each subtype. 2. The presence or absence of active inflammation is not indicated. There is no information about the state of the oral cavity. 3. Is there a difference between smokers and non-smokers? Is there a difference between smokers and nonsmokers in the control group? 4. Figure 5b shows outliers that are not enclosed by an ellipse. What is it and what is it connected with? Where does the number of dots on the diagram come from if 15/15 people are included in the study? 

Author Response

We thank the reviewer for the corrections, suggestions, and questions. 

You will find the detailed point-by-point answers in the attached file, and the modifications applied to the manuscript in the attached revised manuscript.

Best regards 

Round 2

Reviewer 3 Report

The authors responded to the comments of the reviewer, made the appropriate changes to the text of the manuscript. In its present form, the article can be recommended for publication.